# Application of Graphene in Tissue Engineering of the Nervous System

**DOI:** 10.3390/ijms23010033

**Published:** 2021-12-21

**Authors:** Karolina Ławkowska, Marta Pokrywczyńska, Krzysztof Koper, Luis Alex Kluth, Tomasz Drewa, Jan Adamowicz

**Affiliations:** 1Department of Regenerative Medicine, Collegium Medicum, Nicolaus Copernicus University, Curie-Skłodowskiej 9, 85-094 Bydgoszcz, Poland; marta.pokrywczynska@interia.pl (M.P.); tadrewa@gmail.com (T.D.); adamowicz.jz@gmail.com (J.A.); 2Department of Clinical Oncology and Nursing, Collegium Medicum, Nicolaus Copernicus University, Curie-Skłodowskiej 9, 85-094 Bydgoszcz, Poland; krzyskoper@gmail.com; 3Department of Urology, University Medical Center Frankfurt a.M., 60590 Frankfurt am Main, Germany; luiskluth@me.com

**Keywords:** graphene, graphene-based nanomaterials, tissue engineering, nervous system, cell culture

## Abstract

Graphene is the thinnest two-dimensional (2D), only one carbon atom thick, but one of the strongest biomaterials. Due to its unique structure, it has many unique properties used in tissue engineering of the nervous system, such as high strength, flexibility, adequate softness, electrical conductivity, antibacterial effect, and the ability to penetrate the blood–brain barrier (BBB). Graphene is also characterized by the possibility of modifications that allow for even wider application and adaptation to cell cultures of specific cells and tissues, both in vitro and in vivo. Moreover, by using the patient’s own cells for cell culture, it will be possible to produce tissues and organs that can be re-transplanted without transplant rejection, the negative effects of taking immunosuppressive drugs, and waiting for an appropriate organ donor.

## 1. Introduction

The dogma of neural science claims that components of neural systems have poor regenerating capacity in terms of reestablishing axonal connections after injury or diseases. Decades of tissue engineering research have changed this so far highly polarized view and created new perspectives showing potential solutions. The application of the tissue engineering paradigm involving cell transplantation and the design of biomaterials replacing native ECM proved that successfully regenerating neural systems is only a matter of time. In order to achieve this ambitious goal, different scaffolds are currently being evaluated for supporting neural tissue regeneration, by offering unique spatial configurations, superior adhesive profiles, or predictable guiding capacity [1]. In terms of restoring neuronal signaling, the biomaterials’ physical properties, such as electrical conductivity, are of the utmost importance, as well as the spatial biomaterial configuration. Over the last several years, graphene has attracted much attention. It is understandable because it brings with it an infinite number of possibilities for use in the world of medicine, engineering, electronics, and even more industries [2]. Even though graphite, which consists of many layers of graphene, surrounds us on a daily basis, the graphene itself was isolated for the first time in 2004 by mechanical exfoliation using the “scotch-tape” method [3]. This approach allowed the creation of 2D graphene flakes and the first physical characterization of this material. In every aspect, graphene revealed unique features: it is the thinnest of all known materials—only one atom thick, it is the strongest material ever measured—the in-plane carbon bond is stronger than the tetrahedral carbon bond in a diamond. At the same time, it is flexible and stretchable, and can be fully bent and stretched up to 20%. It has the highest thermal conductivity of all materials [4]. All of the above-mentioned properties allow this material to be advantageous for biomedical applications. Although carbon derivatives are often used in biotechnology, graphene is still in the early translational research phase, and its potential is being gradually uncovered [5]. The major strategy so far, to use graphene in tissue engineering, is concentrating on suspending graphene in culture media or applying it as a scaffold for cell culturing. These first attempts are slowly evolving into a new stage of graphene research, focusing on utilizing graphene’s conductivity profile to create signaling cellular interfaces. Scaffolds containing graphene-based nanomaterials (GBNs) can induce neuronal differentiation due to porosity or wrinkles. Additionally, GBNs used as electrodes combined with scaffolds can efficiently induce neuronal differentiation by electrical stimulation. This possibility opens up the prospect of transforming low differentiated cells of the patient into specialized cells that will allow the regeneration of damaged structures of CNS or PNS. Tissue engineering is still developing technology to produce “off-the-shelf organs” for patients in need of transplantation (Figure 1). One of the crucial problems to be solved is to obtain non-toxic, biocompatible material [6,7] that supports differentiation into neural cells to replace intramural neuronal networks. The major limitation of current tissue engineering technology is the impaired function of regenerated organs due to insufficient reconstruction of the neuronal component. Tissue engineering of the nervous system involves regeneration of damaged connections and nerve cells, the creation of functional neuronal networks, and targeted drug delivery. In addition, graphene, due to its unique properties of thinness and conductivity, is enabling the next generation of biotechnology aimed at developing customized scaffolds for neural-machine interfaces or neuroprosthetic devices. Current treatment of injured peripheral nerves is based on surgical reconstruction aimed at joining two nerve endings. Performing mechanical anastomosis, often using autografting, does not restore full functionality. It entails also a number of limitations, including, among others, the fact that the nerve gap cannot be longer than 4 cm, and that still does not always allow 100% restoration of the pre-injury functionality [8]. Damage to the CNS often causes more serious consequences and, hence, is more difficult to repair due to the complicated structure involving the conventional neuronal network and uncontrolled formation of glial scars. To solve the problem of the lack of a support for brain tissue regeneration, it is crucial to develop a biomaterial that will not only be a site for cell adhesion, but also provide the best conditions for cell proliferation and differentiation [9]. The aim of this study was to assess and review the properties, advantages, and limitations of graphene used in tissue engineering of the nervous system.

## 2. Environment of Tissue Regeneration

The healing environment of the central nervous system, in contrast to the peripheral nervous system, is particularly unfavorable for spontaneous regeneration. One of the explanations for this difference is the presence of inhibiting proteins such as myelin-associated proteins (MAP), Nogo proteins, on the surface of the myelin sheath on the outgrowths of axon. These proteins were detected mainly within CNS of adult mammals. They are major components of the glial scar. Rapid formation of the glial scar within CNS damage results in an impermeable barrier for regenerating axons, as glial cells do not conduct neuronal impulses. [10]. Consequently, the formation of glial scars prevents restoration of disrupted neuronal networks. The limitations of current therapeutic approaches to damage to the PNS and CNS have driven many new studies aimed at developing new tissue engineering strategies for the nervous system with the utilization of biomaterials that would allow for differentiation as well as conductivity [11]. Although there is a possibility of a transplant or surgical repair of the interrupted nerve continuity, the nerve gap cannot be larger than 1–4 cm according to the literature. Even though a surgical procedure can be successfully and properly carried out, a transplant does not guarantee normal functionality. Graphene offers the possibility of developing customized conductive scaffolds for neural-machine interfaces or neuroprosthetic devices [8]. In this approach, because of current technological limitations, it could be easier to repair a neural network with graphene than to recreate it de novo. 

## 3. Graphene Characteristics

Graphene aroused great interest with properties that distinguish it from other nanomaterials, such as high biocompatibility, good electrical and thermal conductivity, flexibility, transparency and mechanical strength. Other advantages of the application of GBNs include hydrophilicity, protein adsorption, cell adhesion, cell growth, antibacterial activity, strength, stiffness, and toughness [12]. The strength of graphene can reach 130 GPa, which is 100 times higher than that of steel [13]. Graphene consists of only one single element: carbon, The fourth, unbound electron is responsible for the thermal and electrical conductivity, the speed of which is 1/300th the speed of light. Graphene derivatives each have slightly different properties that allow them to be used potentially in various industries, which is why graphene and GBNs are so popular and so much research is being carried out. For example, the graphene film itself, due to its strength and flexibility, can be used as a material to produce flexible electrical devices. Smart devices that adapt to the patient’s body will allow for even better health protection by monitoring abnormal parameters before the occurrence of a serious, life-threatening disease or death, allowing disease prevention or reducing the harmfulness of treatment [14]. Moreover, GBNs have antibacterial properties, which is a huge advantage for the use of GBNs in tissue engineering of the nervous system [15]. The two-dimensionality and flexibility of graphene and its derivatives also allow for the encapsulation of molecules. The material most commonly used for this purpose is reduced graphene oxide (rGO) due to its better hydrophilic properties. The negative rGO charge prevents the formation of aggregates, which is a constantly bothersome problem for other nanomaterials to overcome [16]. The goal of tissue engineering is to regenerate damaged areas by using a biocompatible scaffold material that should correspond to the physiological conditions as much as possible. Graphene and GBNs are excellent biomaterials with a wide range of properties that enable neuronal differentiation by mimicking the conditions of the extracellular environment of the nervous tissue. GBNs have many wrinkles and ripples on their surfaces that provide good conditions for proliferation and differentiation [17]. An important parameter in cell culture is also the selection of a scaffold with an appropriate hardness; for example, graphene is quite a soft substrate, which in turn is a suitable environment for nervous tissue, while other tissues may prefer harder substrates for proper proliferation and differentiation [18]. Due to the fact that graphene has excellent electrical conductivity, it supports culturing of electroactive cells, i.e., cells of the nervous system [19]. In addition, it is characterized by high specific surface area, low cytotoxicity, and high biocompatibility. Graphene is a special material because it has high modifiability; it is possible to obtain many different graphene-based scaffolds that will mimic in vivo conditions, for example, by changing the number and size of pores, ripples, hydrophilic properties, elasticity, or stiffness, and therefore, is suitable for differentiation in all cell lines [20]. The application of GBNs in tissue engineering of the nervous system also allows drug delivery on demand, crossing the BBB depending on the need [21], reaching damaged areas, and monitoring the activity of cells and molecules in real time [22]. Graphene is also used in neuro-oncology for tumor imaging or delivery of anti-cancer drugs directly to the tumor site [23,24]. Graphene is highly hydrophobic, which prevents its dissolution in aqueous solutions. This is the reason why hydrogels containing graphene are so difficult to obtain. In this situation, graphene is more commonly applied in derivative forms such as graphene oxide (GO) or rGO for cell culturing that suppress the strong hydrophobic properties of unmodified graphene [25]. Due to the possibility of modifying graphene, it is possible to obtain many GBNs such as rGO microfiber, graphene-based polyacrylamide hydrogels (AMGXs), nanoparticles of graphene (NPG), thermally reduced graphene (TRG), single-layered graphene (SG) or multi-layered graphene (MG), printed graphene nanoplatelets (GNPs), graphene foam, graphene film, carbon nanotubes (CNTs), fluorinated graphene, and many more. Table 1, Table 2, Table 3, Table 4, Table 5 and Table 6 list relevant studies that utilize GBN. 

## 4. Neural Interface

In recent years, neural interfaces have become more and more popular, as they allow for the study of basic interactions of the brain-neural system. Furthermore, they are used to treat many neurophysiologic disorders [58]. This has been made possible by significant developments in neuroscience and nanotechnology [59]. A very important aspect of in vivo neural interfaces is their long-term stability [60]. In addition, wireless nerve stimulation and wireless recording of nerve signals allow for the stimulation of even deeply located autonomic nerves in the body, for example, by using technology such as near-field communication [61]. An example of the use of such a method is restoring the function of the bladder, which has lost its normal functionality due to illness or an accident [62]. 

### Graphene: A Neural Interface

The nervous system is composed of a network of nerve cells that receive, process, and react to stimuli from outside and inside the body, allowing other systems to synchronize by generating action potentials at synapses [63]. This mechanism is possible because neurons are electrically excitable and are able to conduct an electrical impulse over long distances. Although most biomaterials show promise for the regeneration of other types of cells, in the case of nerve cells, they may only allow low cell proliferation and stimulate mainly glial scar formation, possibly due to poor scaffold-cell interactions. Therefore, the optimal scaffold designed for neuronal regeneration should be electrically conductive so as to promote the growth of signal transmitting neurons [64,65,66]. Graphene and GBNs allow active participation in neural cell maturation [39]. The use of GBNs as an interface makes it possible to monitor the activity of cells. Moreover, conductive GBNs permit active cell stimulation such as cell-to-cell interaction even between cells that are dispersed [67]. This is of utmost importance for tissue-engineered constructs with cell densities significantly lower than native tissues. Graphene-based scaffolds support the development, branching, and maturation of nerve synapses. They also similarly stimulate the elongation of neurons and formation of extensive neural networks [48]. 

Lu et al. produced a 3D porous graphene coating that successfully and simultaneously recorded electrophysiological records and optical imaging [68]. This is probably due to the unique property of the 3D porous graphene structure that allows the formation of a large specific surface area and superior electrical conductivity, thanks to which a graphene coating can improve the performance of the electrochemical properties of neural electrodes [69].

## 5. Two-Dimensional Graphene-Based Scaffolds

Two-dimensional scaffolds support formation of neural cell monolayer, allowing study of the basic mechanisms of neurons and synapses. [70]. Research on the influence of graphene on the behavior of peripheral neuron PC12 cells and primary DRG neurons by Domenica Convertino et al. showed that 5 days after seeding, PC12 cells plated on graphene were characterized by 27% longer neurite length than cells plated on non-graphene medium. DRG cells also exhibited a positive effect of graphene; in this case, cells plated on both graphene-containing and non-graphene media showed a similar degree of differentiation. The conducted research also demonstrated that graphene can potentially be used as a substance that stimulates the conduction of nerve impulses, showing better properties than gold. The better conductivity was demonstrated by improved elongation of neurons [44]. In order to be able to use graphene in tissue engineering of the nervous system, long-term studies are required to confirm the positive effect of graphene on neuronal differentiation and low cytotoxicity. The team of Sung Young Park et al. investigated the effect of hNSC differentiation on graphene film-coated glass in long-term culturing. Only 10 h after cell seeding, they noticed that the cells on the graphene film-coated glass had better adhesion than cells on the glass. However, after 5 days, the proliferation on graphene film-coated glass was the same as on glass containing growth factors. In further studies by Sung Young Park et al., the team no longer used growth factors, and initiated the differentiation of hNSCs by changing media free of growth factors. Three days after cell seeding, they could not discriminate between cells growing on scaffold containing graphene film and graphene-free scaffolds. Interestingly, after 3 weeks the differences were noticeable. The area with the graphene film was evenly covered with cells that showed a state of very high confluence with neurite outgrowths, while the part without graphene had much less evenly spaced cells, probably due to detachment from the substrate due to differentiation and low adhesion. One month after cell seeding, cells developed a characteristic elongated shape and grew better on graphene in comparison to the glass alone. Immunostaining for the markers GFAP and TUJ1 showed that on graphene film-coated glass, there were almost twice as many neurons as glia. Nevertheless, the results on glass showed the opposite proportions. To confirm the obtained results, the influence of electrical stimulation on differentiation and simultaneous monitoring of neural activity was also investigated using a graphene film as electrode. The cells were stained with Fluo-4-AM to check the calcium concentration in the cells, then a series of electrical pulses were generated. Accordingly, cells showed an average 60–70% increase in fluorescence intensity. However, it should be considered that depending on their position on the electrode, the cells could show different fluorescence intensity, and the glia do not respond to electrical impulses at all. The study confirmed a positive effect on the long-term differentiation of hNSC towards neurons than glia and confirms the positive effect of electrical stimulation on neural differentiation [47]. PC12 cells, i.e., rat pheochromocytoma cells, are widely used in tissue engineering research of the nervous system because they behave similarly to mature dopaminergic neurons (DAs) [71]. These cells are applied as experimental models for in vitro studies on mechanisms related to Alzheimer’s and Parkinson’s disease. Suck Won Hong et al. reported that PC12 cells cultured on glass coverslip with FBS-covered graphene were able to differentiate spontaneously into neuronal cells without adding specific factors. WST-8 test showed that the cells on the FBS-covered graphene glass achieved a higher proliferation rate than the cells grown on the glass without FBS-covered graphene. According to the authors, the difference was significant already after 3 days of culture, and after 7 days, the proliferation on the FBS-covered graphene glass was practically twice as high [56]. The team of Jangho Kim et al. compared the behavior of human mesenchymal stem cells (hMSCs) plated on glass and on glass covered with an SG graphene film. With the addition of normal growth medium only 14 h after plating, the unattached cells were removed using phosphate buffered saline. The team observed that on the graphene-free scaffold, the cells took on a characteristic shape and adhesion, while the scaffold containing the graphene film was characterized by the presence of 3D hMSC spheroids, the size of which ranged from 50–150 µm. The presence was observed for a week, after which the spheroids disappeared; some of them under the influence of extracellular matrix (ECM) produced by the spheroids and the added normal growth medium migrated to the scaffold and showed a characteristic hMSC morphology. This phenomenon was most likely achieved thanks to the high-quality homogeneous monolayer of graphene, which contributed to the development of stronger binding interactions between individual cells than the adhesion of cells to the scaffold. Nevertheless, further research did not detect a difference in morphology between single cells on the scaffold covered with graphene and those on the glass without graphene. However, Western blot analysis demonstrated higher expression of connexin 43 in cells grown on a graphene-containing scaffold, which explained the formation of spheroids. Connexin 43 is a pivotal protein involved in intercellular communication. The aim of the study was to promote neuronal differentiation; therefore, in the next stage a neuronal differentiation medium was added to hMSCs cultured on a graphene scaffold and on glass scaffolds (Figure 2). Only 6 h after this stage of enhanced activation, neuronal differentiation within cells grown on the graphene became remarkable. After 5 days, similar confluence and neuronal overgrowth were observed on both scaffolds, but neurons on the graphene were characterized by more elongation. The author hypothesized that these differences mediated better communication between cells located further away, and could result in the formation of more efficient neuronal networks [45]. 

The team of Chaejeong Heo et al. examined the effect of graphene/polyethylene terephthalate (PET) film electrostimulation on the behavior of Sh-sy5y cells. They reported that the optimal effects were observed after stimulation with a weak electric field, exactly 4.5 mV/mm, with 10 s pulses and a total electrostimulation time of 32 min. The applied technique allowed for increasing the integration between cells and positively influenced the biosynthesis of proteins responsible for cell adhesion and migration [46]. NSCs are multipotent cells that are not only capable of self-renewal, but also differentiation into various nerve and glial cells of the nervous system. NSCs have the ability to proliferate vigorously like all stem cells, but what makes them different is their ability to remain dormant for a long time, providing a reserve of cells in case of a need for regeneration. In addition, the migration capacity allows them to be properly transported to the selected place [72]. However, due to the lack of an appropriate scaffold for neuronal differentiation, brain regeneration consists mainly of the formation of glia. Glia have a number of important functions, such as creating the myelin sheath, allowing for saltatory conduction of the nerve impulse, mediating the nutrition of nerve cells, and can perform defensive functions due to their ability to phagocytose. However, they do not directly conduct nerve impulses [73].

### GO and rGO

Other examples of scaffolds used to regenerate nervous system tissues are GOs and rGOs. GOs are a derivative of graphene monolayer with many groups containing oxygen atoms such as hydroxyl and epoxy functional groups, which are located on the surface of the GOs, and carboxyl groups that appear on the edges of the GO flake. The GO oxidation state enables obtaining substrates with appropriate physico-chemical properties for cell culturing [74]. Thanks to the presence of functional groups, the GO and rGO scaffolds gain hydrophilicity and water solubility. These features create the possibility to penetrate through cell membranes; thus, GOs and rGOs could become effective molecule carriers targeting neural cells. [75]. It must be underlined, however, that care should be exercised because the tensile strength may decrease as the degree of oxidation increases due to cracking of the carbon network. Graphene oxide can then be reduced to rGO, which has physical and chemical properties very close to those of two-dimensional graphene sheet (Figure 3). One of the more important properties of rGOs is their electrical conductivity, which is even higher than that of GOs. rGO differs from GO by the lack of epoxy functional groups on the surface of the rGO flake [76].

Qin Tu et al. examined the growth and branching of primary rat hippocampal neurons on coverslips coated with the chemically functionalized GOs with different surface charges on a scaffold. The team reported that the hippocampal neuron grew well on all GOs (GO-COOH, GO-OCH_3_, GO-NH_2_, GO-PABS) and showed that neuron viability was as high as 96% after 7 days of culturing. Neurons displayed a regular morphology and were arranged and spatially configurated in visible complex networks. However, the study showed that by modifying GO with either negatively or positively charged functional groups, it was possible to induce neuronal differentiation followed by cell maturation. Culturing on a positively charged GO-NH2 scaffold was characterized by the longest maximum length, size, and number of neurite branches, and proved to be slightly better for the maturation of these cells (Figure 4), indicating that positively charged graphene oxide is the best for proliferation and branching of neurites [29].

Adipose derived stem cells (ADSCs), due to their frequent occurrence in vivo, are a good source of cells for research; therefore, they can be a good alternative to MSCs or NSCs, as research shows that under the influence of certain factors, it is possible to differentiate them into nerve cells [77]. Therefore, the team of Zhang-Qi Feng et al. investigated the effect of graphene-based mats on ADSC differentiation into nerve cells. The evaluated scaffold consisted of both GO and rGO. Direct microscopy on day 7 of cell culture showed that ADSCs, which initially exhibited a fibroblast-like shape, acquired a spindle-like shape over time. An MTT test revealed slightly lower cell viability on the scaffold containing GO and rGO than on culture plates (TCPs). Furthermore, there was a noticeable difference in viability between cells cultivated on scaffolds with different GO and rGO proportions. Specifically, cells seeded on scaffolds with higher GO content exhibited increased viability. This implied that GO was characterized by better biocompatibility and lower cytotoxicity than rGO. Immunofluorescence analysis confirmed the beneficial effect of using GO as a scaffold for neuronal differentiation, due to the accelerated neuronal differentiation of cells plated on the scaffold containing graphene compared with the control [78]. Sandr Sánchez-González’s team also compared the effects of GO and rGO on neuronal differentiation, but in this study, rGO turned out to be better than GO. The study compared the effects of GO and rGO in PCL membranes on the behavior and differentiation of human neural progenitor cells. During cell culturing, the cells showed normal adhesion and proliferation. However, the cell population grown on scaffolds containing rGO exhibited more advanced differentiation features and also showed increased length of neurons [79]. The effects of a hybrid of GO and L-theanine (TH) combined with biodegradable poly lactic-co-glycolic acid (PLGA) (PLGA/GO-TH) on the behavior of NSC cells were investigated by Zhiping Qi et.al. They reported that PLGA/GO-TH has better hydrophilic properties than PLGA due to the presence of numerous hydroxyl groups on the GO surface and the hydrophilic properties of L-theanine, thus allowing better cell adhesion and proliferation. The tested cell survival rate did not show a significant advantage of PLGA/GO-TH over PLGA/GO, but comparing these two scaffolds to PLGA alone, a significantly higher cell survival rate was observed on scaffolds containing graphene. Another important parameter was the spread of NSC on the PLGA/GO-TH film; 24 h after seeding, the cells were comparably widespread on all scaffolds. However, 4 and 6 days after seeding, it was reported that NSC cells on the scaffold containing graphene had a significantly higher rate of spreading, but that PLGA/GO-TH film showed the best results. To accurately assess the neuronal differentiation, cells were subjected to both quantitative and qualitative analysis using PCR and immunofluorescence, assuming Tuj-1 expression as a characteristic marker for neurons and that of GFAP for astrocytes. Both tests showed comparable results, indicating a certain tendency: namely, the clearly greatest differentiation of NSC to neurons was characteristic for the PLGA/GO-TH scaffold, followed by PLGA/GO, with comparatively the lowest differentiation observed in cells seeded on PLGA and glass. On the other hand, a completely opposite tendency was shown by differentiation of NSC to astrocytes, where glass and PLGA showed the highest values and, respectively, PLGA/GO and PLGA/GO-TH, the lowest, thus confirming the positive properties of graphene on cell proliferation and differentiation [32].

## 6. Role of Scaffold Dimensionality on Cell Behavior

Two-dimensional cell cultures are very often used in cell research due to the ease of preparation, visibility of single cells, or ease of measurement, but because it is not a natural environment for cells, the results need to be carefully interpreted [80]. A list of 2D graphene-based scaffolds is shown in Table 7. In general, 3D cell cultures better reflect the physiological conditions of the in vitro grown cells, allowing for development of proper proliferation, adhesion, differentiation, migration, protein synthesis, or apoptosis. A list of 3D graphene-based scaffolds is shown in Table 8. Additionally, 3D cultures, due to their spatial configuration, possess porosity and increased surface area for adhesion and cell-to-cell and cell-scaffold communication on a scaffold. Furthermore, 3D cultures, due to their three-dimensionality, allow for the creation of connections between distant neurons, increasing the rate of cell proliferation and differentiation. In comparison, 2D cultures are deprived of the possibility of creating three-dimensionality and, therefore, connections between distant neurons. As Ulloa Severino FP et al. reported, both 2D and 3D cultures promote horizontal growth; however, only 3D cultures facilitate vertical proliferation, allowing for better connections of neurons. The 3D graphene-based scaffolds include composites, foams, fibers and hydrogels [81]. The effects of cell culture dimensionality on cell behavior are listed in Table 9. 

Stem cells are low differentiated cells that can differentiate into their own lineage, but also into other cell lines, which allows them to be widely used in tissue engineering of the nervous system. Stem cells are located in niches, in which there is a state of hypoxia that reduces the risk of damage, in particular of genetic material. The most common place for the collection of stem cells is adipose tissue due to its high presence in the body, bone marrow, and the placenta [88]. In the case of standard organ transplants from a donor, the recipient must take immunosuppressive drugs to reduce the likelihood of transplant rejection; although organ transplantation is a life-saving procedure, the long-term use of immunosuppressive drugs can damage the kidney and liver and lead to the formation of stomach ulcers, cancer, and more frequent infections caused by viruses, bacteria, and fungi. By using the patient’s own cells, such as stem cells to grow the organs needed for transplantation, the likelihood of rejection is extremely low, eliminating the need for immunosuppressive drugs and the associated side effects. In order to grow organs from the patient’s own cells and the individual cells that make up the tissues of this organ, it is necessary to use scaffolds on which cells can proliferate and differentiate. Importantly, 3D scaffolds allow for spatial cell cultures and the formation of cell aggregates, which, due to their complex structure, allow for more reliable research results [89].

**Table 9 ijms-23-00033-t009:** Effects of cell culture dimensionality on cell behavior.

Compared Characteristics	2D Cell Culture	3D Cell Culture	References
Cell shape	– cells are flat, distorted, and have a stretched structure due to strong adhesion of all cells to the scaffold	– cells form 3D structures that reflect the spatial structure of cells in vivo	[90]
Communication	– only cell-scaffold communications	– both cell-to-cell and cell-scaffold communication	[81]
Visibility (analysis of the obtained results)	– each cell can be observed separately, thus allowing better analysis of the results	– difficulties in observing cell morphology	[91]
Cell differentiation	– cells are much less differentiated	– the three-dimensionality of the cell culture allows for better cell differentiation	[92]
Mimicking in vivo conditions	– poor mimic of conditions compared to those in vivo	– good mimic of the extracellular matrix – conditions very similar to those in vivo – have the potential to limit the number of animals used during the early phases of research	[93]
Ability to receive substances from the medium and study of the therapeutic effect of drugs	– the monolayer of the cultured cells results in an even uptake of components from the culture medium by all the cells of the culture – therefore, most drug-related studies have different therapeutic effects in vivo	– ability to take substances from the culture medium differs for all cells – limiting the absorption of drugs from the medium into the cells allows for more reliable results of drug research	[93]
The length of the cell culture and the ability to reproduce the culture conditions	– better for long-term cell cultures – better reproducibility of results	– harder to maintain – more difficult to reproduce the results (one of the exceptions is bioprinting, which is characterized by high repeatability)	[94]
The cost and difficulty of carrying out cell culture	– they typically require easier and cheaper equipment and less experience due to the improved availability of developed cell culture procedures	– they usually require more specialized and more expensive equipment due to the higher demands of 3D cell cultures	[95]
Apoptosis	– factors inducing apoptosis evenly reach all cells from medium, thus increasing the apoptotic process dependent on substances taken from the medium	– access to apoptotic factors is restricted to certain cells, thus reducing the apoptotic process dependent on substances taken from the medium	[96]
Proliferation	– cells proliferate very rapidly	– cells proliferate at a natural rate depending on the conditions	[97]
Cell junction	– less common	– cell junctions are common and allow for cell-to-cell communication	[98]

## 7. 3D Graphene-Based Scaffolds

Scaffolds need particular features to direct the fusion of cells during regeneration in a specific manner adjusted to tissue type. The spatial configuration of the scaffold and its stability play pivotal roles in this process. Therefore, all 3D scaffolds mimic the in vivo conditions in a more efficient way. This is primarily essential for neuronal regeneration, when the outcome is the result of histological structure restoration and reestablishment of signaling within defined pathways. For this purpose, the regenerating scaffold must support the universal regenerating process and at the same time guide elongating axons, which interfere with each other. 

### 7.1. Graphene Foam

Graphene foams are 3D scaffolds containing monolayer graphene, which, due to its porosity and high surface area [99] to volume ratio, mimics the natural microenvironment. It provides long-term support for cell proliferation and guaranties efficient exchange of nutrients and discharge of metabolites. It must be underlined that the size of the pores is important for regeneration purposes. The pores with a diameter of 100–750 μm are optimal, but the pores of larger diameter induce the formation of a pseudo-2D environment, which adversely affects the maturation of nerve cells [100,101]. Graphene foams are also characterized by a large wrinkled structure, which promotes adhesion and neural differentiation corresponding to neural tissue architecture. Relying on animal cells, in this case, mouse NSCs, the research conducted by Ning Li et al. showed very promising results and demonstrated differences between three-dimensional graphene foam (3D-GF) and two-dimensional graphene foam (2D-GF). They reported that mouse NSCs adhered spontaneously to the 3D-GF medium 10 h after seeding. After 5 days, the NSC cells had developed a neural network, which was examined with the use of high-resolution SEM imaging. Additionally, immunofluorescence staining showed that cells also grew inside the scaffold, indicating restoration of 3D configurated tissue in vitro. In the conducted study, the cytotoxicity assays of 2D and 3D graphene scaffold produced comparable results; cytotoxicity was slightly lower with the 3D scaffold, but on the other hand, the value after 5 days indicated that 90% of the cells were alive, highlighting the favorable biocompatibility of graphene. Proliferation was measured by the expression of Ki-67 protein, which suggested that the proliferation of cells on 3D-GF was greater than when using a two-dimensional graphene film. This result may indicate an advantage of 3D-GF over 2D-GF in terms of allowing more cells to be obtained for transplantation. Immunofluorescence staining performed after 5 days demonstrated that in 2D cultures, the expression of neuroepithelial stem cell protein (nestin) was significantly higher than in 3D cultures, but neuron-specific class III beta-tubulin (Tuj-1) and glial fibrillary acidic protein (GFAP) expression were 2.5 and 1.5 times higher, respectively, in 3D cultures than in 2D cultures. This indicates that the 3D cultures on the scaffold graphene enhance the differentiation of neural stem cells into neurons and astrocytes. Cyclic voltammetry assay showed operational potential changes ranging from -0.3 V to 1.0 V, which proved that the current was conducted in the form of interfacial discharges and, hence, could be used for nerve stimulation, as there were no chemical changes in the tissue or electrode. The study indicated that 3G-GF was a better electric stimulator than the 2D graphene film for seeded cells [39]. N. Li et al. compared the electrically active graphene foam with a 2D scaffold culture model for hESC cultures, but their results suggested that there was no significant difference in the dimensions of the culture. However, they confirmed that neuronal cells grown on the graphene scaffold exhibited increased proliferative activity and improved ability to form networks [102,103]. Regeneration of DA has been described by Nishat Tasnim et al., who reported that coating graphene with collagen gel did not affect the basic properties of graphene, but improved its hydrophilicity and beneficial effects on cell differentiation and proliferation. The important results they obtained concern the comparison of the average length of neurite elongation sown on collagen-coated graphene foam and collagen gel, confirming that graphene foam, thanks to its unique properties, allows DA neurons to maintain their morphology and functionality, thus allowing it to be used in the regeneration of the nervous system. The comparison also indicated that it was possible to differentiate MSCs on different scaffolds, but the longest length of neurons was observed on collagen-coated graphene [38]. Akhavan et.al. investigated the effect of differentiation of hNSC into neurons on graphene oxide foam (GOF) using electrical stimulation. They reported that resistance of the GOF was low enough to generate stable electrical stimulation. In addition, results showed that chemically exfoliated GO under UV irradiation resulted in partial deoxygenation of GO as well as the rolling of the GOFs, resulting in even stronger hydrophilic properties, so ultimately it was the rolled GOFs that were used as scaffolds for cell differentiation. Fluorescence staining 72 h after plating showed that cells proliferated both on the surface and within the GOF scaffold and that proliferation was uniform. The study showed that cell proliferation on GOF scaffolds was significantly greater than in the control sample, where cells were seeded on the polydimethylsiloxane (PDMS) commonly used in tissue engineering. However, cells appeared predominantly on the surface of unrolled GOFs compared to the cross section of rolled GOFs. Quantitative analysis of hNSC differentiation indicated that the number of nuclei, ratio of neural cells/ nuclei, and ratio of neurons to glia were all higher on the surface of unrolled GOFs. Nevertheless, electrical stimulation induced better results compared to unstimulated, on cross sections of both the rolled GOFs and the unrolled GOFs [30]. The team of Qinqin Ma et al. compared the effect of 3D-GF stiffness on the maturation of neural stem cells (Figure 5). The soft scaffold showed 30 kPa of the elasticity module and stiff 64 kPa, and both were synthesized using chemical vapor deposition. The team also pointed out that the distance between the stiff scaffolds was smaller than in the case of soft scaffolds, the size of the rigid pores was from 30 μm to 60 μm, and the soft pore size was from 75 μm to 130 μm. The MTT assay did not show any significant differences between the two scaffolds, but 5-bromo-2′-deoxyuridine (BrdU) staining, which determined the rate of proliferation based on Ki67 expression, indicated that stiff scaffold was characterized by 28.54% higher proliferation than the soft one. Cell differentiation after 7 days was examined and indicated that soft scaffolds favored differentiation into neurons and stiff scaffolds into astrocytes [87].

### 7.2. Hydrogels

Hydrogels are 3D structures that accumulate a large amount of water or biological fluids, allowing the creation of realistic tissue structures. The swelling capacity can be related to temperature, pH, light, magnetic field, electric charge, or antigen response, allowing, for example, the control of drug release. Most of the hydrogels are not very durable; the addition of graphene increases their mechanical strength and improves their biocompatibility, allowing the culturing of nerve cells and, thus, the possibility of using hydrogels in tissue engineering [87]. Hydrogels can act as a drug reservoirs or tissue barriers; their release of cells and solutes when needed will enable repair processes without exposure to the storage of these substances in tissues due to continuous secretion [104]. There are two types of gel cross-linking, either chemical or physical. Hydrogels can also be divided according to the origin of the polymers they are made of: natural and synthetic. The more natural the polymer, the better its biocompatibility and the better it mimics in vivo conditions, but an important advantage of synthetic polymers is the predictability of the structure, which allows for mass production [105]. Cristina Martín et al. compared cell culturing in three different media: traditional 2D culture, graphene hydrogels (graphene concentration of just 0.2 mg mL^−1^) and graphene-free hydrogels. They reported that cells seeded on graphene-free hydrogels did not stick to the scaffold and showed very little growth, although the test was repeated 3 times, whereas hydrogels enriched with small amounts of graphene allowed for proper adhesion and maturation of neurons and glia [50]. Accordingly, the team of Krishnangsu Pradhan et al. investigated the effect of choline-functionalized injectable GO (CFGO) hydrogel on brain injury regeneration. The 14-day study showed excellent complete biocompatibility of CFGO; in addition, the study confirmed the increased expression of neuronal markers of cells on CFGO, which demonstrates the beneficial effect of CFGO on the neuronal differentiation of the scaffold. CFGO increased the formation of cytoskeleton fibers that mediated proper communication between cells and took part in cell migration. Additionally, after less than a week of treatment, significant positive effects of the treatment were observed in the rodents under study (male pathogen-free C57BL/6 mice (8−11 weeks)) [106]. Guicai Li et al. conducted a study evaluating the effect of hydrogel enriched with GO/ polyacrylamide (PAM) on SC adhesion and proliferation. The study showed that cells proliferated efficiently on all media; pure PAM hydrogels and the hydrogel with the highest concentration of graphene oxide (1.2%) tended to slow down the proliferation rate over time. On the other hand, the rest of the cells seeded on GO/PAM hydrogels were characterized by normal proliferation throughout the study. GO/PAM hydrogels with GO concentrations of 0.4% and 0.6% demonstrated the best proliferation rates and balanced distributions of culture. Importantly, cells on all other hydrogels, irrespective of the level of proliferation, were documented to show increased adherence. Schwann cells migrated within all GO/PAM hydrogels, preferably at 0.4% GO concentration. In contrast, 90% of the area was covered by cells within 5 days at 1.2% GO concentration. The team observed that cells were significantly smaller than in other GO/PAM hydrogel cultures, and cells on pure PAM hydrogel were characterized by the weakest immersion. They also reported several GO/PAM properties that affect the biocompatibility of the hydrogel. The higher the concentration of GO, the stronger the hydrophobic properties, the higher the mechanical strength, the higher the equilibrium swelling ratio, and the darker the color of the hydrogel. However, the concentration of graphene oxide had no effect on hydrogel degradation in various media. Despite the invisible pores on the surface of graphene oxide (0.2%; 0.8%; 1.2%), GO/PAM hydrogels had a porous structure that may enable the transport of nutrients and metabolites and allow for proper cell proliferation and differentiation [27].

### 7.3. Bioprinting

Three-dimensional bioprinting is a novel technology with a high reproducibility rate, which is definitely desirable because it allows the repeatability of scaffold fabrication for in vitro and in vivo applications. Another advantage is the precise modifiability of scaffold structure, e.g., the thickness or slope of the print to obtain the desired design. Sanjairaj Vijayavenkatarama et al. developed a polycaprolactone (PCL)/rGO scaffold bioprinting method using electrohydrodynamic (EHD) jet technology, which enabled structural modifications of the scaffold, for instance, the number and size of pores, as well as the diameter of the fibers. Graphene oxide was chosen not by accident because it gives the scaffold softness, which is needed for good maturation of nerve cells. The differences in the behavior of PC12 cells on PCL/rGO and on PCL scaffold alone were investigated, and it was shown that cell proliferation and differentiation was higher on PCL/rGO scaffolds than on scaffolds without rGO [35]. A different approach to 3D scaffold production was presented by Yun Qian et al. using the layer-by-layer casting (LBLC) method; they produced and compared scaffolds containing SG or MG and PCL. Some samples also contained polydopamine (PDA) and arginylglycylaspartic acid (RGD). Primary scaffold SG or MG/PCL cytotoxicity studies on Schwann cells were performed at concentrations of 0.1%, 0.5%, 1%, 2%, and 4% SG and MG. Concentrations of 2% and 4% exhibited the highest cytotoxicity of both SG and MG. The proliferation in 1% concentration was better than in the lower concentrations; therefore, for further studies, single and multilayered graphene was used at a concentration of 1%. As a result, a concentration of 1% showed the best biocompatibility. The CCK8 assay performed after 24 h showed no significant differences between the various scaffolds, but changes started to be observed after 120 h and were confirmed in the test after 168 h, showing that PDA/RGD-SG/PCL and PDA/RGD-MG/PCL cells had the best viability. Adhesion to scaffolds was tested by immunofluorescence of adhesion-related proteins: N-cadherin and vinculin. The highest expression was again demonstrated by PDA/RGD-SG/PCL and PDA/RGD-MG/PCL. The proliferation rate was assessed on the basis of Brdu and Ki67 expression, indicating that the highest result was obtained by the culture on PDA/RGD-SG/PCL, followed by PDA/RGD-MG/PCL, while the cultures sown on the medium without graphene showed the lowest level of Ki67 expression. The level of neuronal differentiation was examined with immunofluorescence tests for GFAP and Tuj1, the expression of which respectively indicated the presence of glial cells and neurons. They reported that GFAP expression on PDA/RGD-SG/PCL was significantly the highest and 7.5 times higher than on PCL alone, 5.4 times higher than on PDA/RGD-PCL, and 3.8 times higher than on PDA/RGD-MG/PCL. Relative expression of Tuj1 was significantly higher on graphene substrates, the highest on PDA/RGD-SG/PCL, then on PDA/RGD-MG/PCL, PDA/RGD/PCL, and the lowest on PCL alone, thus confirming that graphene substrates were superior scaffolds for maturation of nerve cells. Further studies, therefore, were carried out in vivo on 90 randomly selected rats, which were then assigned to six groups. Two of them, PDA/RGD-SG/PCL and PDA/RGD-MG/PCL, were Schwann cell-loaded, while the others, PDA/RGD-MG/PCL, PDA/RGD-SG/PCL, PDA/RGD-MG/PCL, and PDA/RGD-PCL, were autograft groups. After 6 and 12 weeks, it was observed that the regeneration of the sciatic nerve was improved on graphene substrates with Schwann cells compared to the others, while still lower than nerve regeneration after autograft. However, after 18 weeks, nerve regeneration on graphene substrates with Schwann cells was comparable to nerve regeneration after autograft [42].

### 7.4. Graphene Fiber

Graphene fiber reflects in vivo conditions due to the possibility of producing very thin layers for cell adhesion. The most common method of producing graphene fibers is the electrospinning method, which allows for the creation of thin fibers that can be easily produced on a larger scale without changing parameters. In addition, traditional graphene fibers can be combined with other graphene derivatives or fiber fillers, giving additional properties that allow better imitation conditions for the extracellular matrix [86,94]. As previously mentioned, scaffolds containing GBNs cannot only induce neuronal differentiation due to porosity or wrinkles, but can directly induce neuronal differentiation using GBNs as electrodes. The big advantages of electrically conductive nanofibers, in addition to the obvious electrical conduction, are also the shape, flexibility, and appropriate softness, which very well mimic the in vivo conditions of neurons. Due to some difficulty in the formation of graphene fiber, the team of Zhang-Qi Feng et al. developed a method for flattening electro-spun poly (vinyl chloride) nanofibers through 2D GO sheets, which allows the production of a uniform GO layer onto nanofibers. The scaffold thus produced was used for culturing neural cells with electrical stimulation, allowing for the generation of correctly shaped motor neurons [107]. Juqing Song et al. developed a method for producing nanofiber scaffolds of PCL with GO using electrospinning technology. They seeded mouse MSCs and low-differentiated PC12 on scaffolds containing different concentrations of GO, noting that mouse MSC proliferation at 24 h was similar on all scaffolds, while low-differentiated PC12 proliferation was highest on graphene-free scaffold and lowest on 1% GO scaffold. After 4 days in both cases, proliferation was significantly lowest on the scaffold with 1% graphene, while after 1 week it was noticed that the greatest proliferation was shown by mouse MSCs cells on 0.3% graphene scaffold, and low-differentiated PC12 cells on a scaffold containing no graphene, but proliferation at 0.1% and 0.3% of GO was practically the same, and 1% GO showed the lowest proliferation in both cases. In addition, after 3 days, field emission scanning electron microscopy (FESEM) images were taken and revealed that mouse MSCs cells on all media showed good adhesion, creating a uniformly coated surface resembling fibroblast morphology, but no changes in cell behavior depending on the concentration of graphene. Low-differentiated PC12 cells were also characterized by good adhesion to the substrate, but in this case, it was possible to distinguish single cells showing a morphology similar to that of neurons, which also merged with other cells, not only with the scaffold [31]. Fluorescence imaging showed that cells cultured from rGO microfiber had a higher density and showed better adhesion to the scaffold than cells cultured from a standard plate, probably due to better protein absorption. Staining also revealed that the cells along the rGO microfiber proliferated evenly, creating networks similar to those formed in natural conditions. Quantitative analysis showed that nestin expression after 3 days was much higher from rGO microfiber than from 2D graphene film, meaning that rGO microfiber scaffold had significantly more immature cells. SEM showed that after 5 days, the confluence was around 50%; another significant change was noticed after a total of 10 days, and the confluence was then around 80%. After the 15th day of culturing, the cells started to build up and showed even greater confluence than on the 10th day. Differentiation was performed after 15 days by immunostaining method; the marker characteristic for neurons was marked with Tuj1, and the protein characteristic for glial cells with GFAP. A significant proportion of cells were Tuj1 positive but a smaller proportion of cells were also GFAP positive; the highest ratio of neurons to glia was shown by rGO microfiber, which showed the highest total number of cells [33]. Weibo Guo et al. reported effects of the rGO microfibers on NSC differentiation. After 3 days, the viability of the sown cells was collected and evaluated sequentially from rGO, a standard cell culture plate, and a 2D graphene film, achieving 95.8%, 94.1%, and 93.5% viability, respectively. The rate of proliferation was also compared using the CCK test at 24, 72, and 120 h after seeding. After 1 day, cell proliferation on all three scaffolds was similar while after the 3rd and 5th days, cells on graphene media showed much better proliferation, whereas the best proliferation was noted on rGO microfiber [36].

## 8. Biodegradation of GBNs

Graphene and its derivatives have attracted significant interest due to their unique properties [108,109,110]. Graphene plays a special role in regenerative medicine, where it could be used for biomedical applications. There are many studies in the literature confirming its positive effect on the proliferation and differentiation of cells, which could then allow the regeneration of tissues or even entire organs. However, an important feature that should be characterized for any biomaterial intended for medical purposes is its biodegradability. A substance that can transform into soluble products that can be processed by the body is considered a biodegradable material. The conducted research indicates that neutrophils and macrophages are directly involved in the biodegradation of GO [111,112], and the resulting products do not adversely affect the functioning of cells [112,113]. It is also worth noting that in water, GO undergoes a constant change in its chemical structure, causing even the cleavage of the bonds between the two carbon atoms. Long-term exposure of GO to water results in the decomposition of the GO flake into structures similar to humic acids, which are the final products of organic material degradation [113,114]. In addition, the degradation of multilayer graphene structures can be facilitated by the naturally occurring hydrogen peroxide in the organism, which, together with the formation and gradual enlargement of defects in the graphene structure, accelerates its degradation [115]. However, it is worth remembering that the biodegradability of each material, including GBNs, is influenced by many factors, such as concentration, dose, duration, model of biological system, and physicochemical characteristics of GBNs, such as size, shape, number of layers, charge, oxidation, and functionalization.

## 9. Conclusions

Graphene is a biomaterial used in many industries, but medical studies are still being conducted on the safety of its use. As presented above, graphene scaffolds could be applied as scaffolds for 2D cultures for research on the basic relationships and behavior of cells. Additionally, graphene also could be applied in 3D cell cultures of cell aggregates or spheroids, which may contribute to reducing the number of animals used for drug testing, thus reducing research costs. Both in vitro and in vivo studies showed great potential for the use of GBNs in the tissue engineering of the nervous system, stimulating the proliferation, adhesion, and neuronal differentiation of cells, allowing not only the repair of damaged nerves caused by injuries, aging, or diseases of the nervous system, but also the production of innervated tissues obtained by tissue engineering techniques, which may revolutionize present-day transplantology. Improving the quality of life of people in need of transplantation, GBNs will significantly reduce the waiting time for transplantation and, thanks to the use of the patient’s own cells for organ reconstruction, will eliminate the impact of immunosuppression, which causes many adverse side effects. Additionally, using only the body’s own cells will help solve the ethical problems related to organ transplantation from deceased donors. Before this becomes possible, it will be necessary to carry out additional studies on the cytotoxicity and biodegradability of graphene scaffolds or to develop a safe method of removing such scaffolds, especially in the case of in vivo use. A significant limitation making comparisons of the results of the studies difficult was the use of graphene of different origin, quality, and purity, which influenced the properties of the tested scaffolds. For this reason, it also will be necessary to develop a method of graphene production that will allow the fabrication of the highest quality graphene and GBNs. At the same time, there is a need to maximally reduce graphene production costs, which would allow mass production and the possibility of using graphene of the same quality worldwide. Another problem is the lack of information as to whether the obtained results were derived from averaged results for all cells of the cell culture, or only from the selected, best-differentiated cell element. Despite many difficulties and inaccuracies in study results, knowledge of the application and properties of graphene has been significantly expanded in recent years thanks to many graphene modification experiments. Obtained graphene derivatives often even better reflect the extracellular conditions of specific tissues, thus facilitating even more effective proliferation and neuronal differentiation of cells and allowing for significant development of tissue engineering of the nervous system. It should also be noted that cells naturally grow in a 3D environment; therefore, 3D scaffolds containing GBNs, due to their unique properties such as porosity and wrinkling, well mimic in vivo conditions, allowing proper cell adhesion, proliferation, differentiation, morphology, and polarity, as well as contact not only cell to cell, but also cell to scaffold. On the other hand, 2D cell cultures are not so accurate due to the formation of an unnatural monolayer, but can be used for basic research and further tests on a larger scale.

## Figures and Tables

**Figure 1 ijms-23-00033-f001:**
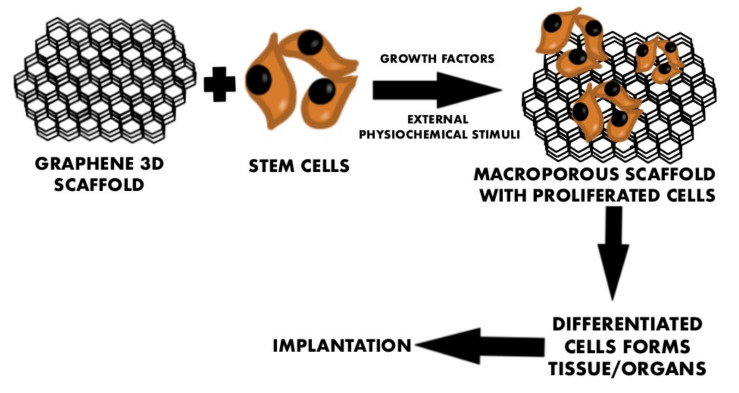
Application of 3D graphene scaffold in tissue engineering of the nervous system.

**Figure 2 ijms-23-00033-f002:**
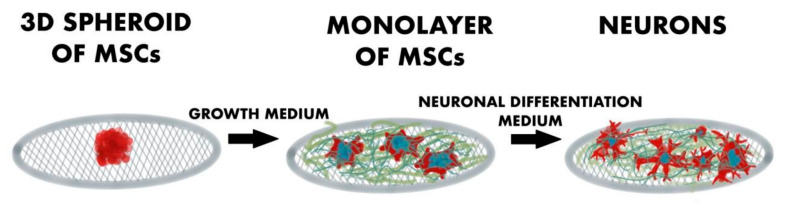
Visual schematic demonstrating the test process of the spheroid formation and neuronal differentiation of hMSCs using high-quality graphene.

**Figure 3 ijms-23-00033-f003:**
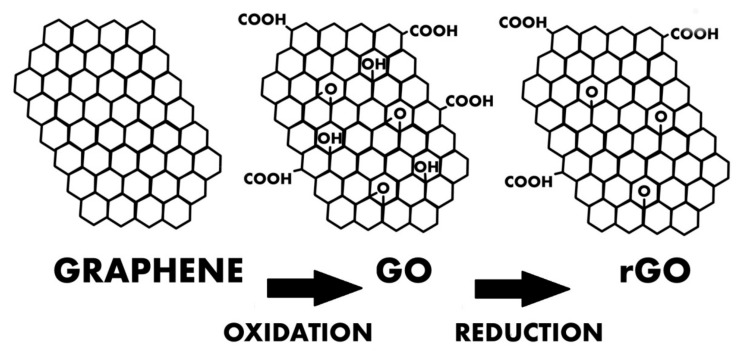
Diagram showing the structure of graphene, GO, and rGO.

**Figure 4 ijms-23-00033-f004:**
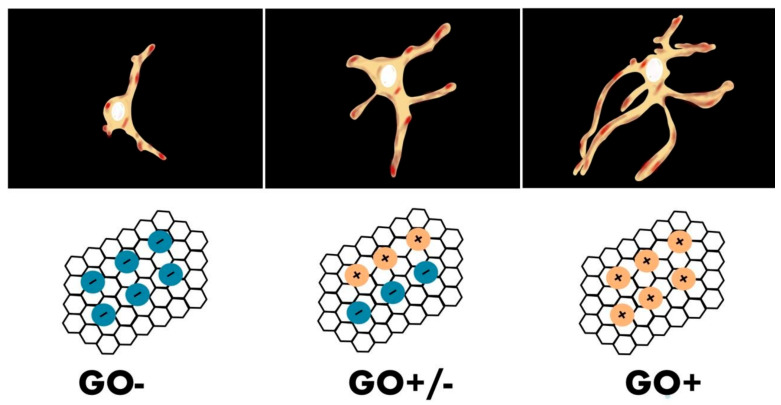
Diagram showing the influence of the scaffold load on the construction of the neuron. Immunochemistry staining images of single hippocampal neurons after 7 days of culture on GO–COOH, GO–OCH3, GO–PABS, and GO–NH2 (from left to right).

**Figure 5 ijms-23-00033-f005:**
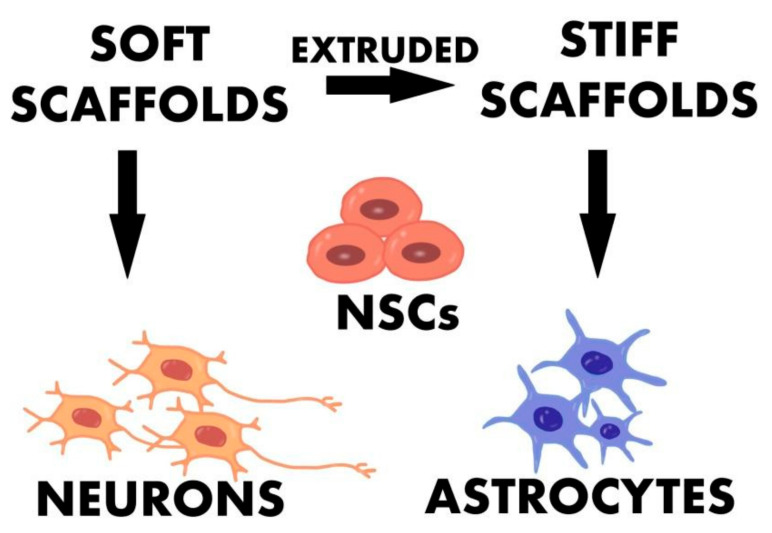
Schematic visualization of relationship between the stiffness of scaffolds and cell differentiation.

**Table 1 ijms-23-00033-t001:** List of relevant studies utilizing GO and rGO.

Types of GBNs Used	Types of Cells Used	Study Performed In Vitro/In Vivo	Year of Publication	Reported Origin of the Graphene	The Most Important Results and Conclusions	Evaluation Methods	References
GO	Rat ENPCs	in vitro	2014	-	14 days were enough to observe differentiated nerve cells	Live/Dead^®^ Viability Kit,	[26]
GO, CNTs, graphene	Mouse ESCs	in vitro	2014	Shandong Tianyuan Co. Ltd. (China)	GO allows for the efficient differentiation of ESCs into dopamine neurons	Immunofluorescence Staining, Real Time PCR (RT-PCR)	[27]
GO	SC	in vitro	2016	-	High concentration of GO is not optimal for the proliferation of SC	CCK-8 Assay, Immunofluorescence Staining, Microscopic Analysis	[28]
GO	Primary Rat Hippocampal Neurons	in vitro	2014	Nanoon (Hebei, China)	Positively charged scaffold (GO-NH2) characterized by the best neuronal proliferation	SEM Imaging, Immunochemistry Staining, Fluorescence Imaging	[29]
GO	hNSC	in vitro	2015	-	Cell proliferation on GOFs was significantly higher than in the control sample where cells were sown on the commonly used in tissue engineering PDMS	Fluorescence Imaging, SEM Imaging	[30]
GO	Mouse MSCs and PC12	in vitro	2015	Sigma Aldrich (USA)	The GO / PCL scaffold allowed for better proliferation and differentiation of mMSCs and PC12-L	Cell Morphologies Using FESEM, CCK-8 Assay, qRT-PCR	[31]
GO	NSCs	in vitro	2020	Chengdu Organic Chemicals Co., Ltd., China	NSC cells on the scaffold containing graphene had the highest rate of spreading	Survival Assays, MTT Assay, RT-PCR	[32]

**Table 2 ijms-23-00033-t002:** List of relevant studies utilizing rGO.

Types of GBNs Used	Types of Cells Used	Study Performed In Vitro/In Vivo	Year of Publication	Reported Origin of the Graphene	The Most Important Results and Conclusions	Evaluation Methods	References
rGO microfiber	Neural Stem Cells (NSCs)	in vitro	2017	-	rGO microfibers may constitute suitable conditions for the cell culture of nerve cells	Immunofluorescence Staining, Fluorescent Calcium Imaging, Quantitative Polymerase Chain Reaction (qPCR)	[33]
rGO	hNSCs	in vitro	2013	-	GO-TiO2 scaffold electrostimulation allowed not only to increase the proliferation of hNSCs but also allowed for neuronal differentiation	Immunofluorescence Staining	[34]
rGO	PC12	in vitro	2018	Sigma-Aldrich Pte Ltd., Singapore	Cell proliferation and differentiation were higher in PCL / rGO scaffolds than in scaffolds without rGO	SEM Imaging, Prestoblue Assay, RT-PCR, Fluorescence Microscopy Imaging	[35]
rGO nanofibers	hMSCs	in vitro	2016	-	From day 5 of culture, the cells on the graphene scaffold showed better proliferation	SEM Imaging, Confocal Microscopy Imaging	[36]

**Table 3 ijms-23-00033-t003:** List of relevant studies conducted with utilization of graphene foam.

Types of GBNs Used	Types of Cells Used	Study Performed In Vitro/In Vivo	Year of Publication	Reported Origin of the Graphene	The Most Important Results and Conclusions	Evaluation Methods	References
graphene foam	Human Embryonic Stem Cell (hESC)	in vitro	2018	Graphene Laboratories, Inc. (Graphene Foam Calverton, NY, USA)	Porous structure of the graphene foam allows for cell penetration onto the scaffold	Quantitative Real-Time Polymerase Chain Reaction (qRT-PCR), Immunofluorescence Staining, SEM Imaging, Helium Ion Microscopy Imaging	[37]
graphene foam	Mesenchymal Stem Cells (MSCs)	in vitro	2018	Graphene Supermarket (Calverton, NY)	Graphene foam allows the differentiation of MSCs into selected cells of the nervous system	Flow Cytometry Analysis	[38]
graphene foam, graphene film	Mouse NSCs	in vitro	2013	-	The graphene scaffold allows good interaction between the scaffold and cells, which is essential for good cell differentiation	SEM Imaging, Cell Viability Assay, Immunofluorescence Staining, Western Blotting	[39]
graphene foam	NSCs	in vitro	2014	-	The study suggests that only 3D graphene foam has antimicrobial properties, while 2D scaffolding does not	Flow Cytometry Analysis, Enzyme-Linked Immunosorbent Assay (ELISA), Western Blotting, Microscopic Analysis, MTT Assay, Immunohistological Staining	[40]

**Table 4 ijms-23-00033-t004:** List of relevant studies conducted with utilization of SG.

Types of GBNs Used	Types of Cells Used	Study Performed In Vitro/In Vivo	Year of Publication	Reported Origin of the Graphene	The Most Important Results and Conclusions	Evaluation Methods	References
SG	Human Brain Vascular Pericyte (HBVP) Cells	in vitro	2016	-	A significantly higher amount of HBVP cells was observed on the scaffold containing graphene	Optical Microscopy Imaging	[41]
SG AND MG	Rat Schwann Cell (rat SC)	in vivo, in vitro	2018	Suzhou Tanfeng Graphene Technology Co., Ltd. (China)	The SG and MG scaffolds allow for the regeneration of damaged peripheral nerves	CCK-8 Assay, SEM Imaging, Immunofluorescence Staining	[42]
SG	Rat Pheochromocytoma (PC12)	in vitro	2016	Neutrino (Iran)	Promising use of the SG and chitin scaffold for the proliferation of nerve cells	MTT Assay	[43]
SG	PC12 and Rat Dorsal Root Ganglion (DRG) primary neurons	in vitro	2018	-	The scaffold containing graphene allowed extending the length of the neurons by 27% compared to the control sample	Viability Assays, Optical Microscopy Imaging	[44]
SG and MG	Rat SC	in vivo	2018	Suzhou Tanfeng Graphene Technology Co., Ltd.	Cultures on PDA/RGD-SG/PCL and PDA/RGD-MG/PCL showed results similar to autograft	CCK-8 Assay, SEM Imaging, Immunofluorescence Staining	[42]
SG	hMSCs	in vitro	2015	-	High quality single-layer graphene (SG) allowed obtaining a spheroid on a 2D scaffold, which lasted 7 days	Western Blotting, Nissle Staining, qRT-PCR Fluorescent Calcium Imaging	[45]
SG	Human Neuroblastoma Cells (Sh-sy5y Cells)	in vitro	2011	-	The best effects were observed with stimulation using a weak electric field	Immunofluorescence Staining Optical and Fluorescence Microscopic Imaging	[46]

**Table 5 ijms-23-00033-t005:** List of relevant studies conducted with utilization of graphene film.

Types of GBNs Used	Types of Cells Used	Study Performed In Vitro/In Vivo	Year of Publication	Reported Origin of the Graphene	The Most Important Results and Conclusions	Evaluation Methods	References
graphene film	Human NSCs (hNSCs)	in vitro	2011	-	Due to its unique properties, graphene allows the differentiation of hNSCs mainly into neurons, not glia	Immunofluorescence Staining, Microarray Experiments	[47]
graphene film	Mouse Hippocampal Cells	in vitro	2011	-	Graphene is a good environment for the development of mouse hippocampal cells; it also allows their neuronal differentiation	Analyzed Via Phase Contrast Microscopy	[48]
graphene film	Mouse NSCs	in vitro	2013	-	Graphene film allows the differentiation of cells that are able to communicate with other cells	Immunofluorescence Staining	[49]

**Table 6 ijms-23-00033-t006:** List of relevant studies conducted with utilization of other GBNs.

Types of GBNs Used	Types of Cells Used	Study Performed In Vitro/In Vivo	Year of Publication	Reported Origin of the Graphene	The Most Important Results and Conclusions	Evaluation Methods	References
AMGXs	Primary Rat Hippocampal Neurons	in vitro	2017	Bay Carbon Inc.	Cell networks between cultured cells were observed only on graphene-containing scaffolds	Immunofluorescence Staining, Fluorescent Calcium Imaging	[50]
NPG	-	in vivo	2017	-	The most myelinated axons were observed on scaffolds containing graphene	Differential Scanning Calorimetry, Fracture Surfaces of The Membranes, Dynamic Mechanical Analysis	[51]
TRG	Mouse NSCs	in vitro	2016	-	Scaffold containing TRG allows for appropriate proliferation and adherence of mouse NSC	Immunofluorescence Staining, Morphological Analysis of Neurons and Oligodendrocytes, Cell Death Assay	[52]
GNPs	Mammalian NE-4C NSC	in vitro	2020	-	The obtained ink creates suitable conditions for the cell culture of nerve cells	Scanning Electron Microscopy (SEM) Imaging	[53]
graphene	Retinal Ganglion Cells (RGCs)	in vitro	2018	-	Despite the lack of significant influence of the use of graphene on cell proliferation, the possible use of graphene as an electrode has been confirmed	Cell Survival Assay, Receptor-Mediated Endocytosis Assay Neurite Outgrowth Assay, Ion Channel Activity Assay	[54]
graphene nanogrids	hNSCs	in vitro	2013	-	Graphene nanogrids, due to their unique properties, allow for neuronal differentiation	Immunofluorescence Staining, Fluorescence Imaging,	[55]
CNTs	PC12	in vitro	2014	-	CNTs showed the best results in cell proliferation of all the materials tested	Immunofluorescence Staining	[56]
Fluorinated graphene	MSCs	in vitro	2012	-	Fluorinated graphene improves the proliferation of MSCs	Immunofluorescence Staining	[57]

**Table 7 ijms-23-00033-t007:** Compilation of 2D graphene-based scaffolds.

	Characteristics	Advantages	Limitations	References
Graphene	– single-layer structure of carbon atoms arranged in the shape of a honeycomb	– good electrical conductor – porous structure	– strong hydrophobic properties – different size of pores depending on the sample	[47,82]
GO	– graphene monolayer with many groups containing oxygen atoms such as hydroxyl and epoxy functional groups	– strong hydrophobic properties are eliminated in favor of hydrophilic properties, which allow GO to be used as a drug carrier and allow penetration of BBB – good electrical conductor – porous structure	– can reduce mechanical strength – different size of pores depending on the sample – presence of functional groups containing oxygen may increase toxicity	[83,84]
rGO	– graphene monolayer with hydroxyl functional groups	– -an even better electrical conductor than GO – porous structure	– can cause bond breakage between adjacent carbon atoms – different size of pores depending on the sample	[75,76]

**Table 8 ijms-23-00033-t008:** Compilation of 3D graphene-based scaffolds.

	Characteristics	Advantages	Limitations	References
Foams	– solid structure foam containing GBNs	– strongly folded structure containing numerous pores, creates a large surface for adhesion, distribution of substances, and cell proliferation	– pores of too large diameter may form, which may induce 2D culture conditions	[85]
Fibers	– thin fibers that can be easily produced on a larger scale without changing parameters	– high repeatability – thin fibers very well mimic in vivo conditions	– depending on the method of obtaining, the strength may differ	[86]
Hydrogels	– 3D scaffold containing graphene, GO, or rGO and large amounts of fluids	– changes in temperature, pH, pressure, magnetic field, or electric charge allow the drug release to be controlled as needed – soft scaffolding suitable for nerve cell proliferation – thanks to the addition of GBNs, the strength of the hydrogel is increased	– they may not be resistant to mechanical damage	[87]
Bioprinting Products	– method that allows the creation of a biomaterial containing both GBNs and living cells	– high repeatability due to the full mechanization of production	– use of high pressure or high temperature may damage the cells	[35]

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
