# Peer review of "Application of Graphene in Tissue Engineering of the Nervous System"

_ijms, 2021, doi:10.3390/ijms23010033_

Round 1

Reviewer 1 Report

Ławkowska' team emphasis the idea that application of graphene in tissue engineering of the nervous system, this paper is not divided clearly, full of grammar error. This paper should be rejected. If your team have solved these following problems, we will send a recommending letter to editor.
1. What is the difference between this review and other  graphene tissue, which one is the focus.
2. Please invite a foreign professor about graphene to revise this paper
3.There exists little discussion on structural modifications, nanometer size regulation and photocatalysis, please add these discussions.

Author Response

  1. What is the difference between this review and other graphene tissue, which one is the focus.???

The aim of the study was to discuss available data about the role of graphene in tissue engineering of neural tissue. To our knowledge it is the first review which addresses this topic. Our article contains and summarizes most of the available research on the use of graphene in tissue engineering of the nervous system. Based upon previously conducted research many tables were constructed to present the advantages, limitations, and the use of various forms of graphene in the clearest possible way.

  1. Please invite a foreign professor about graphene to revise this paper.

Due to the limited time given for revision we couldn’t invite independent professor to review the article. However, I would like to emphasize it politely that one of our authors is native English speaker and he once more revised the article to improve it grammatically.

  1. There exists little discussion on structural modifications, nanometer size regulation and photocatalysis, please add these discussions.

We agree with the reviewer that suggested topics are worth to be discussed. Nevertheless, the review is limited by the number of pages and inclusion of these paragraphs would make it too long. Moreover the photocatalysis in graphene research isn’t related to tissue engineering studies. Our review is dedicated mainly to researcher working of regenerative medicine with biological backup rather than of those who work on graphene fabrication per se. Moreover, there is not enough data on photocatalysis in order to include it in this review. 

Reviewer 2 Report

The paper can be published in present form

Author Response

Thank you very much for your review.

Reviewer 3 Report

This paper tentatively reviews the status of graphene in tissue engineering of the nervous system. There has been an increasing number of studies showing that combining graphene with other materials to form nanocomposites can provide exceptional platforms for both stimulating neural stem cell adhesion, proliferation, differentiation and neural regeneration. This suggests that graphene nanocomposites are greatly beneficial in neural regenerative medicine. However, note that this topic has been treated in recent reviews (not cited here) by Bei et al. Molecules (Basel) 2019, 24, 658 and by Maleki et al. BioMol Concepts 2020, 11, 182-200.

To be suitable for publication in IJMS, the authors must show the novelty of their review article. Additional issues should be addressed as follows.

  1. The objective of this review article should be described in the introduction.
  2. The bibliography is incomplete. Please consider the following articles: Akhavan, Mater. Chem. B, 2016,4, 3169-3190 and references within; Zhang et al., J. Nanomaterials 2020, Article ID 2519105; Ku, et al., Advanced Healthcare Materials, 2013, 2, 244–260; Chung et al., Accounts of chemical research2012, 46, 2211–2224; Yang, et al. Chemical Society Reviews 2013, 42, 530–547; Yang et al. Materials Today, 2013, 16, 365–373; Park et al. Nat. Commun. 2014, 5, 6258; Ku et al. Adv. Healthc. Mater. 2013, 2, 244–260; Xu et al. ACS Nano 2016, 10, 3267–3281.
  3. (line 46) please provide a reference.
  4. Advantages of GBNs should be specified in Section 3.
  5. Authors provide six tables which are not called in the main text. Content of these table must be discussed in the related section.
  6. Description of graphene as a neural interface (Section 4) is rather maigre. Characteristics of materials given in references should be discussed.
  7. A comparison of GBNs to other novel materials which excel at neural tissue engineering should presented.
  8. The possibility of biodegradation of graphene-based materials should be evoked.
  9. Figure and Table captions are not satisfactory. Figures are stand-alone sections. Therefore, they should provide all the required information in the captions such that the reader is not required to refer to the main text.
  10. Each section of the main text should be organized in several paragraphs for easy reading.
  11. References should be written in journal style (see instructions to authors).
  12. In conclusion: Reconsider after major revision.

Author Response

  1. The objective of this review article should be described in the introduction.

The remark was applied in the article.

  1. The bibliography is incomplete. Please consider the following articles: Akhavan, Mater. Chem. B, 2016,4, 3169-3190 and references within; Zhang et al., J. Nanomaterials 2020, Article ID 2519105; Ku, et al., Advanced Healthcare Materials, 2013, 2, 244–260; Chung et al., Accounts of chemical research2012, 46, 2211–2224; Yang, et al. Chemical Society Reviews 2013, 42, 530–547; Yang et al. Materials Today, 2013, 16, 365–373; Park et al. Nat. Commun. 2014, 5, 6258; Ku et al. Adv. Healthc. Mater. 2013, 2, 244–260; Xu et al. ACS Nano 2016, 10, 3267–3281.

The remark was applied in the article, the following articles were included in this review.

  1. (line 46) please provide a reference.

The remark was applied in the article.

  1. Advantages of GBNs should be specified in Section 3.

Advantages of GBNs are specified in Section 3 and discussed in the rest of the paper.

  1. Authors provide six tables which are not called in the main text. Content of these table must be discussed in the related section.

The remark was applied in the article.

  1. Description of graphene as a neural interface (Section 4) is rather maigre. Characteristics of materials given in references should be discussed.

The remark was applied in the article.

  1. A comparison of GBNs to other novel materials which excel at neural tissue engineering should presented.

This is a very interesting idea, but our goal was to present the advantages and disadvantages only of GBNs

  1. The possibility of biodegradation of graphene-based materials should be evoked.

There are not enough conducted studies that show relevant results on the possibility of biodegradation of graphene-based materials. This problem is mentioned in the summary of the review in order to draw attention to the problem that needs to be further explored.

  1. Figure and Table captions are not satisfactory. Figures are stand-alone sections. Therefore, they should provide all the required information in the captions such that the reader is not required to refer to the main text.

The remark was applied in the article.

  1. Each section of the main text should be organized in several paragraphs for easy reading.

I can’t agree with reviewer that the article isn’t properly organized. The separate topics are already discussed in paragraphs. In our opinion, further splitting of paragraphs would significantly reduce the clarity of the article. However we revised the article in this regard and slightly  rebuild it.

  1. References should be written in journal style (see instructions to authors).

The remark was applied in the article.

  1. In conclusion: Reconsider after major revision.

Reviewer 4 Report

The manuscript by Ławkowska et al. “Application of graphene in tissue engineering of the nervous system” is interesting and notworthy. It requires revision to address major concerns.

Comments.

  1. The authors should cross-verify the uses of abbreviations in the text.
  2. In the introduction section, various facts and descriptions or pieces of information have been presented without citations. The appropriate citations should be provided for such various statements. 
  3. Lines 65-67, the various biocompatible materials such as polyhydroxyalkanoates are well known either directly or in their modified forms with nanomaterials. Thus, the potential advantages and disadvantages of graphene-based materials should be stated over other such materials for the significant purpose of this article i.e. doi: 10.1007/978-981-13-3759-8_10, doi: 10.1016/j.biortech.2021.124737.
  4. Tables 1-6, some recent studies should be added 2020-2021. The year of publication entry should be deleted. The authors need to provide more quantitative information on each Table such as the concentration of materials used and how much repair? Also, include such details in the text.
  5. A better illustration should be provided to highlight the significance and summary of the article.

Author Response

  1. The authors should cross-verify the uses of abbreviations in the text.
  2. In the introduction section, various facts and descriptions or pieces of information have been presented without citations. The appropriate citations should be provided for such various statements.
  3. Lines 65-67, the various biocompatible materials such as polyhydroxyalkanoates are well known either directly or in their modified forms with nanomaterials. Thus, the potential advantages and disadvantages of graphene-based materials should be stated over other such materials for the significant purpose of this article i.e. doi: 10.1007/978-981-13-3759-8_10, doi: 10.1016/j.biortech.2021.124737.
  4. Tables 1-6, some recent studies should be added 2020-2021. The year of publication entry should be deleted. The authors need to provide more quantitative information on each Table such as the concentration of materials used and how much repair? Also, include such details in the text.

If such data were provided in the study, they were included in the table, as it was written in the summary, the lack of concentration is a big problem that makes all of the results presented not comparative.

  1. A better illustration should be provided to highlight the significance and summary of the article.

All of the above remarks, were included in the revision of the paper.

Round 2

Reviewer 1 Report

Dear editor,

Hi, Ławkowska's work has emphasis on application of graphene in tissue engineering of the nervous system.  There exists a great deal of problems, if your team solve those following questions, we will send a recommending letter to editor.

  1. This paper is full of grammar error, please invite foreign researcher to revise.
  2. It should be including much more details about graphene structure and nanocomposite.
  3. what about cells for cell culture, and the corresponding cell culture mechanism

Author Response

  1. This paper is full of grammar error, please invite foreign researcher to revise.

Due to the limited time given for revision, we couldn’t invite an independent professor to review the article. However, I would like to emphasize it politely that one of our authors is a native English speaker and he once more revised the article to improve it grammatically.

  1. It should be including much more details about graphene structure and nanocomposite.

Minor changes were included in the work, for example, in chapter 3 (line 115), but I believe that there are already enough articles focused on details about graphene structure and nanocomposite and it is unnecessary to include all the information in this review.

  1. What about cells for cell culture, and the corresponding cell culture mechanism 

The information on cells used for the tests is included in tables 1-6 and in the text. Additionally, information was included (lines 401-413) that, when using one's own stem cells, it is possible to culture tissues or even organs that are almost 100% likely to receive, and the side effects of administration of immunosuppressive drugs are minimized: „Stem cells are low differentiated cells that can differentiate into their own lineage, but also into other cell lines, which allows them to be widely used in tissue engineering of the nervous system. Stem cells are located in the niches of stem cells, in which there is a state of hypoxia, that reduce the risk of their damage, in particular of genetic material. The most common place for the collection of stem cells is adipose tissue due to its high presence in the body, bone marrow and the placenta [87]. In the case of standard organ transplants from a donor, the recipient must take immunosuppressive drugs to reduce the likelihood of transplant rejection, although organ transplantation is a life-saving procedure, long-term use of immunosuppressive drugs can damage kidney, liver, lead to the formation of the stomach ulcers, cancer and lead to more frequent infections caused by viruses, bacteria and fungi. By using the patient's own cells, such as stem cells to grow the organs needed for transplantation, the likelihood of rejection is extremely low, eliminating the need for immunosuppressive drugs and the associated side effects.”

Reviewer 3 Report

The amended version of this manuscript is not satisfactory. The revision is very brief and does not answer the reviewer's questions.

I recommend major revision. The authors should address the following issues.

  1. First the replied letter to reviewer is not satisfactory. For each action, the authors must describe the modification with the corresponding location (line number) in the main text.
  2. The authors must justify the review article in the front of the recent reviews of this topic treated by However, by Bei et al. Molecules (Basel) 2019, 24, 658 and by Maleki et al. BioMol Concepts 2020, 11, 182-200. These works should be cited.
  1. Point #1: I don’t see the objective of this review article in the introduction.
  1. Point #2: Despite the claim “The remark was applied in the article” the references are not included. Page number is missing in [1].
  2. Point #4: the advantages of GBNs should be specified in Section 3.
  3. Point #5: Content of tables 1-6 are not discussed in the text.
  4. Point #8: where is the paragraph treating the biodegradation?
  5. Point #11: despite the claim, the references’ style is unchanged.
  6. Etc.

Author Response

  1. First the replied letter to reviewer is not satisfactory. For each action, the authors must describe the modification with the corresponding location (line number) in the main text.

Point #1.         The objective of this review article should be described in the introduction.

The remark was applied in the article. (lines 84-86)

Point #2.         The bibliography is incomplete. Please consider the following articles:

 Akhavan, Mater. Chem. B, 2016,4, 3169-3190 and references within;(line 1)

 Zhang et al., J. Nanomaterials 2020, Article ID 2519105; (line 115)

 Ku, et al., Advanced Healthcare Materials, 2013, 2, 244–260;(line 58)

Chung et al., Accounts of chemical research 2012, 46, 2211–2224;(line 151)

Yang et al. Materials Today, 2013, 16, 365–373;(table 7; line 399)

Park et al. Nat. Commun. 2014, 5, 6258;(line 176)

The remark was applied in the article, the following articles were included in this review.

Point #3.         (line 46) please provide a reference.

The remark was applied in the article.(line 44)

Point #4.         Advantages of GBNs should be specified in Section 3.

Advantages of GBNs are specified in Section 3 (lines 110-115) and discussed in the rest of the paper.

Point #5.         Authors provide six tables which are not called in the main text. Content of these table must be discussed in the related section.

The remark was applied in the article. Tables 1-6 contain information about all the studies described throughout the text. Table 7 contains a text reference (line 385). Table 8 contains a text reference (line 388). Table 9 contains a reference in the text (line 398).

Point #6.         Description of graphene as a neural interface (Section 4) is rather maigre. Characteristics of materials given in references should be discussed.

The remark was applied in the article. (lines 172-205)

Point #8: where is the paragraph treating the biodegradation?

The remark was applied in the article.(lines 667-689)

Point #11: despite the claim, the references’ style is unchanged.

References were formatted by Section Managing Editor with a note quoting: "Thanks for submitting to IJMS. We have made a layout of your manuscript according to our journal format. Please check and confirm (Only need to check the content, not the format). "

  1. The authors must justify the review article in the front of the recent reviews of this topic treated by

         The work was verified and all recommendations were taken into account.

  1. However, by Bei et al. Molecules (Basel) 2019, 24, 658 and by Maleki et al. BioMol Concepts 2020, 11, 182-200. These works should be cited.

          The remark was applied in the article.(line 436 and 311)

Reviewer 4 Report

The authors have revised the manuscript appropriately. The manuscript can be accepted as it.

Author Response

Thank you very much for your review.

Round 3

Reviewer 3 Report

This revised manuscript (3rd version) is not worthy to be published in IJMS. I might think that the authors refuse to make the requested changes

  1. In the response letter to the reviewer, the authors did not describe the changes made to the main text but simply say "The remark was applied in the article" which avoids a relevant reply. How can I judge the change?
  2. In the introduction, the novelty of this review article compared to previous ones [by Bei et al. Molecules (Basel) 2019, 24, 658 and by Maleki et al. BioMol Concepts 2020, 11, 182-200.] is not justified.
  3. Contents of Tables 1-6 are not discussed in the main text. The authors simply wrote “List of relevant studies that utilize GBN are shown in Tables 1-6.”
  4. Despite the claim, the references’ style is unchanged. According the “Instructions for Authors” a reference of journal article should be organized as follows: [reference number] Author 1, A.B.; Author 2, C.D. Title of the article. Abbreviated Journal NameYearVolume, page range.
  5. Many references are incomplete. [1] page number missing; [13] article ID is missing; [59] there is no page number but article ID which is missing; Etc.